# A High-Frequency Temporal-Interference Alternative Current Stimulation Device Using Pulse Amplitude Modulation with Push–Pull Current Sources

**DOI:** 10.3390/bioengineering12020164

**Published:** 2025-02-08

**Authors:** Jia-Hao Bai, Szu-Chi Huang, Po-Lei Lee, Kuo-Kai Shyu, Chao-Jen Huang, Tsung-Chih Chen, Sheng-Ji Lai

**Affiliations:** 1Department of Electrical Engineering, National Central University, Taoyuan 32001, Taiwan; bai790902@gmail.com (J.-H.B.); r310102930719@gmail.com (S.-C.H.); kkshyu@ee.ncu.edu.tw (K.-K.S.); a5531749@gmail.com (C.-J.H.); audi.tc.chen@gmail.com (T.-C.C.); sjilai2001@gmail.com (S.-J.L.); 2Department of Medical Research, Cathay General Hospital, Taipei 10630, Taiwan

**Keywords:** temporal interference, TI, alternating current stimulation, pulse amplitude-modulation, PAM-TI, push–pull circuit

## Abstract

This study proposes a high-frequency Pulse Amplitude-Modulation Temporal-Interference (PAM-TI) current stimulation device, which utilizes two sets of Amplitude-modulated transcranial alternating current stimulation (AM-tACS): one AM frequency at f0 (where f0 = 2 kHz) (source 1) and the other AM frequency at f1 = f0 + △f (where f1 = 2.01 kHz) (source 2), to generate a △f (where △f = 10 Hz) envelope modulated at a fc (where fc = 100 kHz) high carrier frequency. The high carrier frequency reduces body impedance and conserves more stimulation power, allowing it to penetrate the skin and reach the subcutaneous region. The proposed PAM-TI technique elevates the two current sources to a 100 kHz carrier frequency. Instead of the challenges associated with generating high-frequency stimulation currents using an MCU and DAC, the proposed PAM-TI stimulation device achieves this by simply utilizing a pair of complementary pulse-width modulations (PWMs). The push–pull technique is employed to balance the charging currents between the anode and cathode, synchronizing the current timing of Source 1 and Source 2 under the fc modulation condition. To minimize signal attenuation, the PAM circuit is integrated directly into the electrode, ensuring the high-frequency signal is generated close to the body and preventing degradation from long wires. Additionally, a dry pin-type spring-loaded electrode is used to reduce interference caused by hair when placed on the head. The device’s validity and current directionality were verified using a scalp tissue-mimicking phantom composed of agar and saline.

## 1. Introduction

At present, deep brain stimulation (DBS) is dominated by invasive brain stimulation technology, which is mainly suitable for brain diseases such as Parkinson’s disease [1], Alzheimer’s disease [2], and epilepsy [3]. Although DBS is helpful for some brain diseases, it can also cause some side effects, such as cognitive impairment, confusion, infection and bleeding during surgery, and risk of complications [4].

Non-invasive transcranial direct current stimulation (tDCS), transcranial alternating current stimulation (tACS) and transcranial magnetic stimulation (TMS) technologies can still only stimulate superficial areas of the brain. In order to achieve the goal of non-invasive deep brain stimulation, Grossman et al. (2017) [5] proposed temporal-interference electrical stimulation (TI). TI uses two or more channels to stimulate high-frequency tACS sinusoidal currents of different frequencies (such as 2 kHz and 2.01 kHz) to reach deeper brain tissues. The electric fields generated by the currents of each channel will meet and interfere with each other in the brain. The interference is an electric field envelope that oscillates at a frequency difference (10 Hz) between the two, and this modulated electric field is used to stimulate deep brain tissue. Since high-frequency current can directly pass through cells and intercellular substance, this high-frequency carrier current can reach deep brain areas (such as thalamus and hippocampus) without damaging the neurons on the conduction path [6,7]. However, most of the current research on TI electrical stimulation waveforms uses 2 kHz and 2.01 kHz for interference, where the bioimpedance is still a few kilohms, resulting in the amplifier being unable to operate at low voltage. According to previous studies [8,9], higher stimulation frequencies can reduce biological impedance, meaning that at the same stimulation current intensity, stimulation signals with higher frequencies can achieve the desired effect with a lower output voltage. The reduction in output voltage not only enhances the safety of the system but also reduces the supply voltage requirements, thereby decreasing the power consumption and heat dissipation of the output operational amplifier (OPA), making it more suitable for wearable system applications. For example, a higher frequency input of 100 kHz can reduce the bioimpedance to about 100 Ω, conserving more stimulation energy to penetrate through the skin surface. As shown in Figure 1, the impedance decreasing trend plot flattens out after 100 kHz.

This study considered the advantages of AM-tACS and high-frequency carriers. We propose PAM-TI to modulate the f0 (where f0 = 2 kHz) and f1 = f0 + △f (where f1 = 2.01 kHz) AM signals using a high carrier fc (where fc = 100 kHz) frequency. By modulating the f0 and f1 frequency waveforms with the fc frequency carrier, the interfered electrical stimulation at △f frequency can be modulated at a high fc frequency and transmitted. This technique avoids the need for traditional high-frequency signal generators that require expensive high-clock MCUs and fast DACs, as seen in previous studies where the TI device implemented used an external DAC [10,11]. This study, however, does not rely on an external DAC. The study adopts a complementary PWM and digital switch architecture to modulate low-frequency f0 and f1 signals with an fc frequency carrier. As shown in Table 1, standard STM32 series microprocessors and DACs cannot output 100 kHz signals, and their minimum adjustable resolution does not meet the requirement of 10 Hz resolution. To address this issue, the proposed architecture does not directly generate two waveforms with a 10 Hz frequency difference at a high modulation frequency. Instead, it generates basic 2 kHz and 2.01 kHz waveforms and then uses PWM technology to produce PAM waveforms. This approach makes the PAM-TI technique more flexible, allowing any low-frequency AM-tACS signal to be elevated to a high-frequency carrier for transmission. Ultimately, we propose a high-frequency PAM-TI stimulation system. Compared to the traditional TI stimulation frequency of 2 kHz, PAM-TI can achieve stimulation frequencies of up to 100 kHz, significantly enhancing the system’s frequency range and application potential. In terms of efficiency and safety, PAM-TI effectively reduces the operating voltage of the system due to the lower output impedance, which not only improves the system’s safety but also reduces power consumption, thereby enhancing overall energy efficiency. This makes it more suitable for prolonged use and wearable applications. In terms of cost, increasing the frequency of traditional TI systems is constrained by the operating frequency of embedded systems, leading to a significant rise in system costs. In contrast, PAM-TI utilizes PWM signals and peripheral circuits for Pulse Amplitude Modulation (PAM), enabling higher-frequency stimulation signals without significantly increasing hardware costs, thus overcoming the technical and cost limitations of traditional TI systems at high-frequency stimulation.

Past studies of electrical stimulation waveforms reported that the usage of biphasic waveforms is an essential safety issue to achieve charge balance during stimulation. It is a safety requirement to minimize tissue damage that is caused by unbalanced charge injection at the tissue–electrode interface [12,13]. This is why the PAM-TI technique proposed in this study employs complementary PWMs to maximize the phase difference between the two current sources in the TI system. Additionally, each current source is paired with a push–pull circuit design to draw current from the anode to the cathode within each source, preventing interference between the two current sources.

In terms of current source circuit design, past research usually used transformers to isolate the signals between distinct electrical stimulation channels [11,14]. However, transformers are bulky and heavy, making them challenging to integrate into wearable devices. In addition, transformers generate annoying noise, which can disturb users during sleep or rest. The current source in this study adopted the improved Howland circuit, in which the load end is isolated through an Operational Amplifier (OPA) buffer circuit and the feedback current was minimized to improve accuracy [15]. The electrical stimulation device designed in our work did not use a transformer for load-end isolation, making it lighter and more suitable for integration into wearable devices.

## 2. Materials and Methods

### 2.1. Electrode-Integrated PAM Circuit and Complementary PWM Control Board

This study aims to modulate two sets of AM-TACS waves at a high carrier frequency. Since high-frequency analog signals may experience significant attenuation over long-distance transmission [16,17], the PAM output should be positioned close to the human body to avoid the attenuation of high-frequency currents. Therefore, this study integrates the PAM circuit into the electrode to minimize the distance between the generation of the high-frequency PAM signal and the stimulation site on the body, effectively preventing the attenuation of high-frequency analog signals in long wires. The electrode-integrated PAM (EIP) circuit receives PWM signals and power supply from the main control board. The main control board includes a complementary PWM generation circuit and a microprocessor with a DAC to produce low-frequency AM-TACS. The low-frequency AM-TACS and complementary PWM signals are transmitted from the main control board, as shown in Figure 2a to the electrode-integrated PAM (EIP electrode), as shown in Figure 2b. Within the EIP electrode, the PAM circuit modulates the AM-TACS signal to a high carrier frequency for subsequent use in temporal-interference (TI) current stimulation.

The EIP circuit designed in this paper offers high compatibility and can be used with either dry or wet electrodes, which can be selected according to the specific application needs, as shown in Figure 2c,d. Previous studies [18,19] have indicated that both dry and wet electrodes can establish effective electrical contact on the skin. Although dry electrodes result in higher contact impedance, they are generally preferred by users since they do not require conductive gels, highlighting their potential for practical applications. In the context of this paper, the PAM-TI stimulation frequency is much higher than that of traditional TI, which further mitigates the impact of higher contact impedance in dry electrodes, ensuring the stability of the stimulation signal and the applicability of the system.

### 2.2. Pulse-Amplitude Modulation Temporal Interference (PAM-TI)

This paper designs a high-frequency PAM-TI waveform, which uses an fc (where fc = 100 kHz) high-frequency carrier to pulse amplitude modulate the f0 (where f0 = 2 kHz) and f1 = f0 + △f (where f1 = 2.01 kHz) sine waves in the traditional TI waveform. Figure 3a shows the traditional TI of a 2 kHz sine wave, which is transformed into Figure 3e using PAM, where the amplitudes of the f0 sine wave are modulated by the fc carrier. Figure 3b shows the traditional TI of a 2.01 kHz sine wave, which is transformed into Figure 3f using PAM, where the amplitudes of the f1 sine wave are modulated by the fc carrier. Figure 3c shows the 10 Hz envelope waveform produced by the interference between the 2 kHz and 2.01 kHz waves in the traditional TI. Figure 3g shows the △f envelope waveform produced by the interference of f0 with the fc PAM and f1 with the fc PAM in the PAM-TI. By comparing Figure 3d and 3h, we can see that the waveform proposed in this paper can shift the 2 kHz and 2.01 kHz components to the higher frequency side of f0 + fc and f1 + fc through the fc PAM carrier.

### 2.3. Circuit Design and Simulation

This section will show the PAM-TI current source generator circuit design and related formula derivation. The following circuit design is drawn and waveform output by LTspice circuit simulation software Version 24.1.1. Figure 4 shows how to generate PAM-TI Source. First, the signal is a sine wave generated by the DAC of the MCU, and the carrier is a square wave generated by the PMW of the MCU. The circuit uses switches to modulate the sine wave and the PWM carrier, and then multiplies the modulated signal by the gain of 2. The final output of the differential amplifier with the gain of 1 (Vref) is the difference between the signal and the multiplied modulated signal.

Figure 5 shows that the green line is the output (Vref) of the PAM-TI Source waveform generator circuit. The blue line is the sine wave generated by the DAC of MCU. Previous studies have shown that modulated waveforms can be transmitted to deep brain regions with less energy [20].

The formula derivation is as follows (1)–(4):(1)$$s\left(t\right)=\sum_{n=-\infty }^{\infty }m\left(nT_{s}\right)h(t-nT_{s})$$

$T_{s}=$ sample period determined by PWM

$m\left(nT_{s}\right)=$ sample value of *m*(*t*) obtained at $t=nT_{s}$ is generated by DAC

$h\left(t\right)=$ Rectangular pulse of unit amplitude and duration

*T* and it is defined as(2)$$h\left(t\right)=\left\{\begin{array}{c} 1,\;0\leq t\leq T\\ 0,\;otherwise \end{array}\right.$$(3)$$K=\frac{R_{p2}}{R_{p1}}=\frac{R_{p4}}{R_{p3}}=1,\;R_{p1}=R_{p2}=5\;kΩ,R_{p3}=R_{p4}=5\;kΩ$$(4)$$V_{ref}=\left(\left(1+\frac{R_{p5}}{R_{p6}}\right)\times S\left(t\right)-DAC\right)\times K,\;R_{p5}=R_{p6}=5\;kΩ,\;K=1$$

#### 2.3.1. Improved Howland Current Pump with Buffer Circuit

The current source circuit used in this article is an improved Howland current pump with buffer as shown in Figure 6, which has the added benefit of minimizing error by practically eliminating $I_{feedback}$ current due to the added buffer [15].

Considering the impact of component errors on the output current accuracy of the Howland current source circuit, the following derivation is performed:

Based on the virtual ground characteristic of the OPA, Equation (5) can be obtained.(5)$$V_{x}=\frac{R_{2}+∆R_{2}}{R_{1}+R_{2}+∆R_{1}+∆R_{2}}V_{ref}+\frac{R_{1}+∆R_{1}}{R_{1}+R_{2}+∆R_{1}+∆R_{2}}V_{o}=\frac{R_{3}+∆R_{3}}{R_{3}+R_{4}+∆R_{3}+∆R_{4}}V_{y}$$
where $∆R_{1}$, $∆R_{2}$, $∆R_{3}$, and $∆R_{4}$ represent the resistance errors of resistors $R_{1}$, $R_{i4}$, $R_{3}$, and $R_{4}$, respectively, and R≫∆R. Simplifying Equation (5) yields Equation (6).(6)$$V_{ref}=\frac{R_{3}+∆R_{3}}{R_{3}+R_{4}+∆R_{3}+∆R_{4}}\times \frac{R_{1}+R_{2}+∆R_{1}+∆R_{2}}{R_{2}+∆R_{2}}V_{y}-\frac{R_{1}+∆R_{1}}{R_{2}+∆R_{2}}V_{o}$$

To simplify the analysis, set $R_{1}=R_{3}$ and $R_{2}=R_{4}$, which gives:(7)$$V_{ref}=\frac{R_{1}+∆R_{3}}{R_{1}+R_{2}+∆R_{3}+∆R_{4}}\times \frac{R_{1}+R_{2}+∆R_{1}+∆R_{2}}{R_{2}+∆R_{2}}V_{y}-\frac{R_{1}+∆R_{1}}{R_{2}+∆R_{2}}V_{o}$$

Equation (7) can be simplified through basic algebraic operations to obtain Equation (8).(8)$$V_{ref}=\frac{R_{1}^{2}+R_{1}R_{2}+∆R_{1}R_{1}+∆R_{2}R_{1}+∆R_{3}\left(R_{1}+R_{2}+∆R_{1}+∆R_{2}\right)}{R_{1}R_{2}+R_{2}^{2}+∆R_{2}\left(R_{1}+R_{2}\right)+\left(∆R_{3}+∆R_{4}\right)\left(R_{2}+∆R_{2}\right)}V_{y}-\frac{R_{1}+∆R_{1}}{R_{2}+∆R_{2}}V_{o}$$

Since $R_{1}^{2}$, $R_{2}^{2}$, and $R_{1}R_{2}$ will be much larger than the error terms, Equation (8) can be approximated as:(9)$$V_{ref}=\frac{R_{1}^{2}+R_{1}R_{2}+∆R_{1}R_{1}+∆R_{2}R_{1}+∆R_{3}\left(R_{1}+R_{2}+∆R_{1}+∆R_{2}\right)}{R_{1}R_{2}+R_{2}^{2}+∆R_{2}\left(R_{1}+R_{2}\right)+\left(∆R_{3}+∆R_{4}\right)\left(R_{2}+∆R_{2}\right)}V_{y}-\frac{R_{1}+∆R_{1}}{R_{2}+∆R_{2}}V_{o}$$

Simplifying (9) gives:(10)$$V_{ref}=\frac{R_{1}}{R_{2}}\left(V_{y}-V_{o}\right)-\frac{∆R_{1}R_{2}-∆R_{2}R_{1}}{R_{2}^{2}+∆R_{2}R_{2}}V_{o}$$

Based on (10), the output current $I_{o}$ is:(11)$$I_{o}=\frac{\left(V_{y}-V_{o}\right)}{R_{s}+∆R_{s}}=\left[\frac{1}{R_{s}}-\frac{∆R_{s}}{R_{s}^{2}+∆R_{s}Rs}\right]\left[\frac{R_{2}}{R_{1}}V_{ref}+\frac{∆R_{1}R_{2}-∆R_{2}R_{1}}{R_{1}R_{2}+R_{1}∆R_{2}}V_{o}\right]$$

Through Equation (11), the output current can be separated into the ideal output current $I_{o_{ideal}}$ and the output current error $I_{o_{tolerance}}$.(12)$$I_{o}={I_{o}}_{ideal}+I_{o_{tolerance}}$$(13)$$I_{o_{ideal}}=\frac{1}{R_{s}}\times \frac{R_{2}}{R_{1}}V_{ref}$$(14)$$I_{o_{tolerance}}=\left(\frac{1}{R_{s}+∆R_{s}}\right)\frac{∆R_{1}R_{2}-∆R_{2}R_{1}}{R_{1}R_{2}+R_{1}∆R_{2}}V_{o}-\frac{∆R_{s}}{R_{s}^{2}+∆R_{s}Rs}\frac{R_{2}}{R_{1}}V_{ref}$$

Based on relation (15), the output current error can be designed according to the application requirements.(15)$$\eta \times I_{o_{ideal}}\geq I_{o_{tolerance}}$$
where η is the maximum acceptable error percentage in the application.

#### 2.3.2. Current Source Circuit OPA Dissipation

The power consumption of the OPA $P_{D}$ can be divided into internal power $P_{D_{internal}}\;and\;output\;power\;P_{D_{output}}$, as expressed by the following equation:(16)$$\left\{\begin{array}{c} P_{D}=P_{D_{internal}}+P_{D_{output}}\\ P_{D_{internal}}=I_{Q}\times V_{s}\\ P_{D_{output}}=I_{o}\times \left(V_{s}-V_{o}\right) \end{array}\right.$$
where $I_{Q}$ is the static current of the OPA, $V_{s}$ is the supply voltage of the OPA, $I_{o}$ is the output current of the OPA, and $V_{o}$ is the output voltage of the OPA. The temperature rise of the OPA component can be estimated using Equation (17).(17)$$T_{J}=T_{A}+P_{D}\times \theta_{JA}$$
where $T_{J}$ is the junction temperature of the OPA, $T_{A}$ is the ambient temperature, and $\theta_{JA}$ is the thermal resistance (°C/W).

Based on Equations (16) and (17), it is evident that PAM-TI has a significant advantage over traditional TI. Due to the higher stimulation frequency of PAM-TI, it effectively reduces the bioimpedance, thereby lowering the operational amplifier (OPA) supply voltage $V_{s}$ and output voltage $V_{o}$. This not only significantly reduces the current source OPA power dissipation $P_{D}$, improving the overall system efficiency, but also results in a lower operating temperature under the same cooling conditions, mitigating the risk of overheating and enhancing the stability during prolonged operation.

#### 2.3.3. Push–Pull Current Source Design

This paper adopts a push–pull current source architecture, using two sets of PAM-TI Source, each of which contains two sets of Howland current sources. In this design, only one MCU is needed to control four sets of Howland current sources, and the MCU’s PWM and PWMc (complementary PWM) are used to perform 100 kHz pulse amplitude modulation and to synchronize each Howland current source, and two sets of DACs are used to generate 2 kHz and 2.01 kHz sine waves, respectively, as shown in Figure 7.

This paper designs a mechanism that can control the current push and pull of the current source and the complementary control method. The following takes the PAM-TI source1 as an example, and the schematic diagram is shown in Figure 7. The MCU can output two sets of symmetrical PMW and PWMc, and through the two sets of DACs of the MCU, DAC1 outputs 2 kHz sinewave and DAC2 outputs 2.01 kHz sinewave, respectively, to generate a PAM-TI waveform. It can be seen from Figure 8 that PWM and PWMc are complementary signals, which also control the current direction of the Howland current source Push I+ and Pull I−.

Based on a comparison between previous literature and this paper, the focus is on charging current balance, no transformer, and finally the supported current stimulation frequency. The comparison of related neural electrical stimulation device papers is summarized in Table 2. It can be seen that this paper proposes charging current balance control, and a transformer-free electrical stimulation current source is used, and the stimulation frequency can reach 100 kHz due to the use of PAM technology (Ahmedabad, India).

## 3. Results

The electrical stimulation circuit proposed in this paper was verified by the nerve equivalent electric circuit proposed by McNeal et al. [26] to observe the waveform of the current and the waveform of interference in the tissue stimulated by our proposed current source. The actual circuit connection diagram is shown in Figure 9. The two pairs of PAM-TI source1 and PAM-TI source2 are connected to the equivalent circuit, and the interference current waveform generated is shown in Figure 9. It can be seen that the interference frequency ∆f = 10 Hz has between 2 kHz and 2.01 kHz current sources.

Finally, this paper adopts the electrical simulant agar of the human scalp tissue-mimicking phantom proposed in the paper of Morales-Quezada et al. [27], as shown in Table 3, and places it in a 7 × 7 matrix sampling box made by 3D printing.

The two PAM-TI sources are placed in the upper left of the sampling box and the lower left for PAM-TI Source1. The upper right and the lower right are PAM-TI Source2. The wiring diagram is shown in Figure 10.

To verify the current directionality of the push–pull current source, PAM-TI Source1 was connected to the upper left/lower left of the sampling box, and PAM-TI Source2 was connected to the upper right/lower right of the sampling box. A probe was then used to record the voltage generated by the current flowing through the tissue-mimicking phantom, allowing us to observe the current flow distribution via the voltage difference. Peak-to-peak voltage (Vpp) measurements were used to generate a heatmap. As shown in Figure 11, a significant voltage difference was observed in the vertical direction, while the voltage difference in the horizontal direction was less pronounced. This experiment aimed to confirm the influence of REF (where only the ground of the measuring probe is connected to the sampling box) or GND (where both the ground of the measuring probe and the stimulator system are connected to the sampling box) on the direction of the current. Therefore, REF or GND was placed at the middle or bottom-center position. It was observed that a slight voltage depression occurred around the reference point or ground, but the vertical and horizontal current directionality remained intact. These observations validate the feasibility of the push–pull current source proposed in this study.

According to the observation results of the previous experiment, the direction of current flows vertically. Therefore, the tissue-mimicking was measured by a differential method to verify the current interference situation. The vertical and the horizontal directions at the center point of the tissue-mimicking phantom can be observed from Figure 12 below. It can be found that there are 10 Hz interference waveforms in both the vertical and horizontal target areas, but after post-processing, the vertical differential voltage amplitude and the 100 kHz carrier wave are more obvious horizontally.

## 4. Discussion

In this paper, we provide a detailed explanation of the design methodology for the electrical stimulation device. We propose utilizing PAM technology to modulate the carrier frequency to 100 kHz, achieving a frequency resolution of 10 Hz without requiring an expensive high-end high-frequency DAC. This approach reduces the impedance of skin tissue, conserving more energy for effective tissue stimulation. The proposed high frequency PAM-TI stimulation device has the following features:Through the PAM architecture, we can modulate low-frequency waveforms (e.g., 2 kHz and 2.01 kHz in this study) to high frequencies (e.g., 100 kHz). This method avoids the use of expensive high-end high-frequency DACs while precisely maintaining the slight frequency difference between the two waveforms (e.g., 10 Hz). This technique utilizes high-frequency modulation to reduce the impedance of skin tissue, thereby conserving more energy for tissue stimulation.This study adopts complementary PWM signals and a push–pull circuit design in the PAM-TI technique to maximize the phase difference between the two current sources in the TI system, ensuring charge balance and minimizing tissue damage during stimulation, thus addressing safety concerns related to unbalanced charge injection at the tissue-electrode interface.To minimize attenuation, the study integrates the PAM circuit directly into the electrode, ensuring that the high-frequency signal is generated close to the body. This approach prevents the degradation of the signal in long wires. Additionally, a dry pin-type spring-loaded electrode is used to overcome the interference caused by hair when placed on the head.

Based on the above characteristics of this device, the PAM-TI current stimulation device was proposed, and the interference waveform and interference frequency were verified through the neural equivalent circuit. Finally, the current directionality of the push–pull current source structure was explored through a scalp tissue phantom made of agar and saline. In this paper, we introduce PAM-TI and analyze its advantages over traditional TI, particularly in increasing stimulation frequency, reducing biological impedance, enhancing system efficiency, and improving heat dissipation. However, the impact of PAM-TI on safety and neural stimulation efficiency requires further validation and evaluation. In future research, we will conduct biological experiments to further assess the effectiveness of PAM-TI in real biological environments and ensure its safety and efficacy for clinical applications.

## Figures and Tables

**Figure 1 bioengineering-12-00164-f001:**
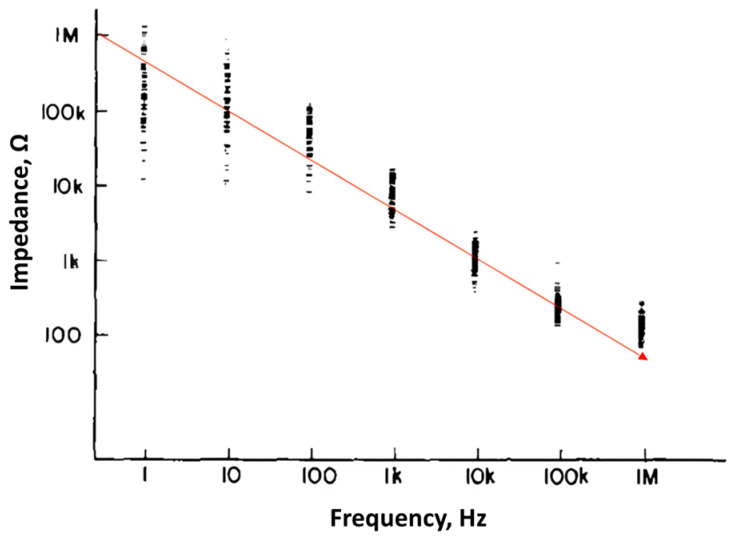
Bioimpedance versus frequency [9].

**Figure 2 bioengineering-12-00164-f002:**
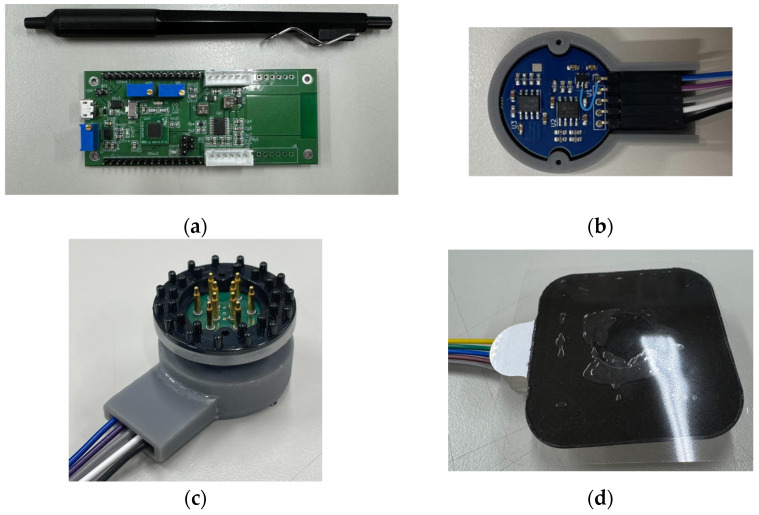
(**a**) Integrated power supply main control board. (**b**) Electrode-integrated PAM circuit. (**c**) EIP circuit with dry pin-type spring-loaded electrode. (**d**) EIP circuit with wet electrode.

**Figure 3 bioengineering-12-00164-f003:**
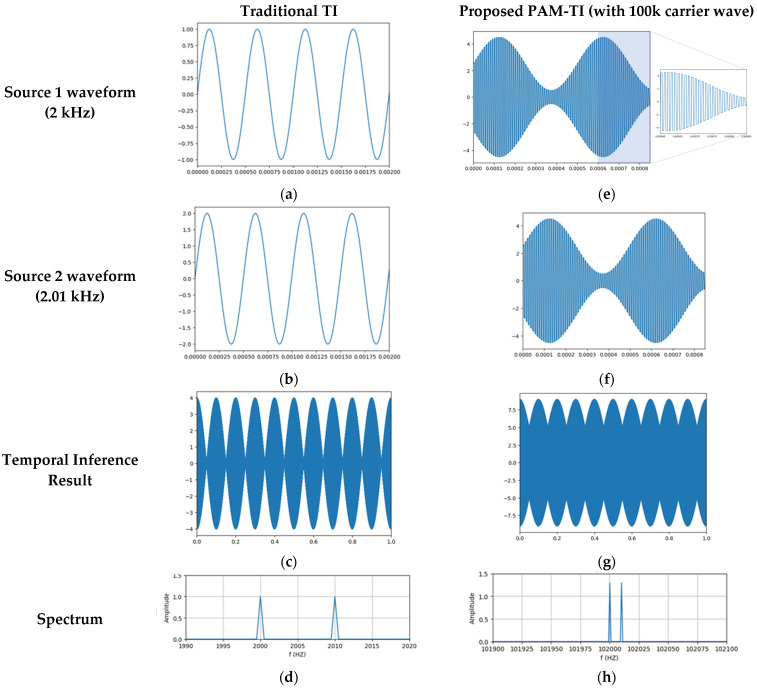
(**a**) The sine wave with frequency 2 kHz; (**b**) The sine wave with frequency 2.01 kHz; (**c**) The TI wave consists of the envelope formed by the superposition of TI Source1 and TI Source2; (**d**) The spectrum of conventional TI; (**e**) The f0 sine wave with fc PAM; (**f**) The f1 sine wave with fc PAM; (**g**) The PAM-TI wave consists of the envelope formed by the superposition of PAM-TI Source1 and PAM-TI Source2; (**h**) The spectrum of PAM-TI.

**Figure 4 bioengineering-12-00164-f004:**
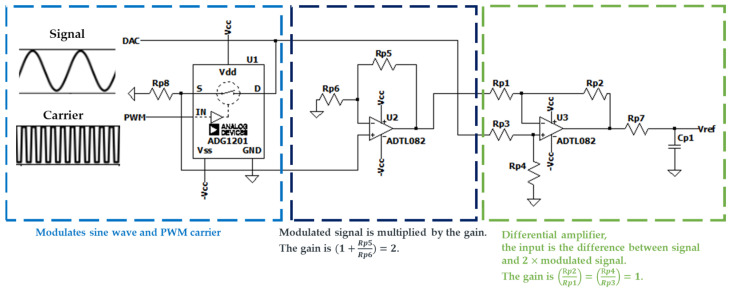
PAM-TI Source waveform generator circuit.

**Figure 5 bioengineering-12-00164-f005:**
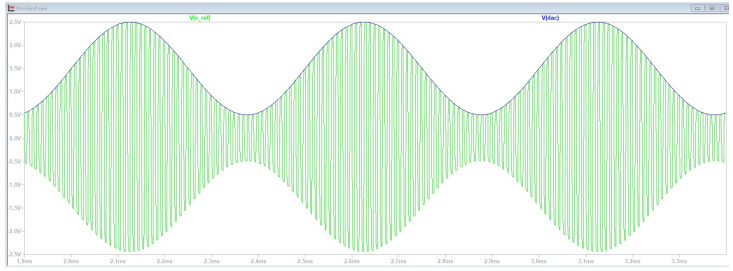
The output (Vref) of PAM-TI Source waveform generator circuit.

**Figure 6 bioengineering-12-00164-f006:**
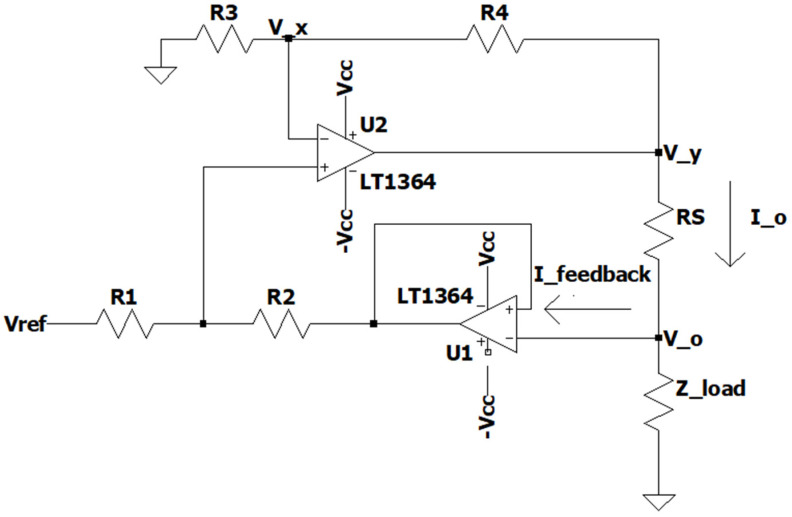
Improved Howland current pump with buffer circuit.

**Figure 7 bioengineering-12-00164-f007:**
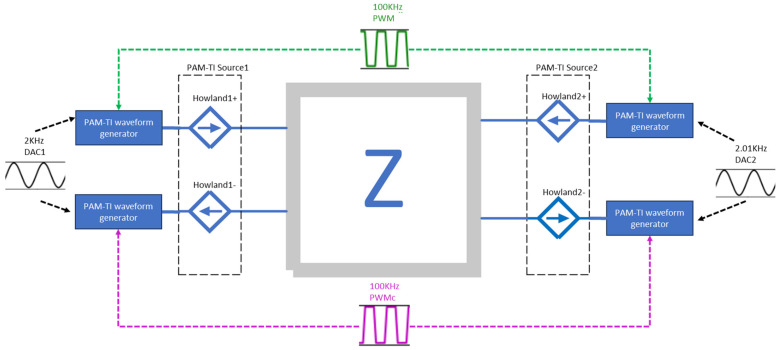
Push–pull current source architecture.

**Figure 8 bioengineering-12-00164-f008:**
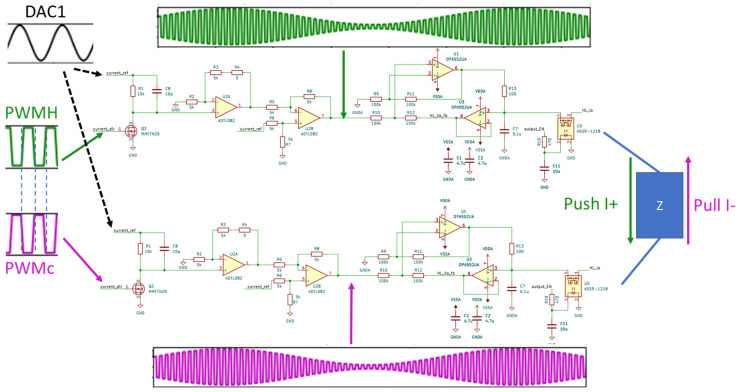
Push–pull current control diagram.

**Figure 9 bioengineering-12-00164-f009:**
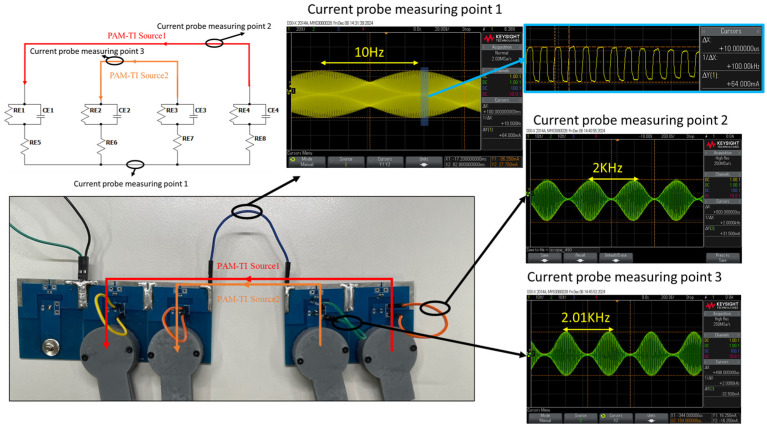
The actual circuit connection diagram.

**Figure 10 bioengineering-12-00164-f010:**
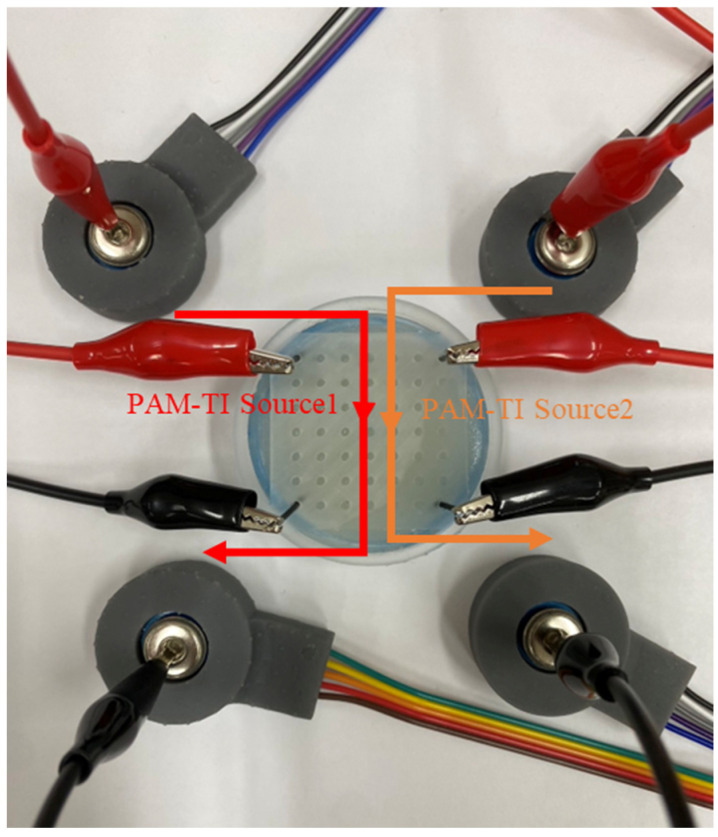
Agar measurement experiment diagram.

**Figure 11 bioengineering-12-00164-f011:**
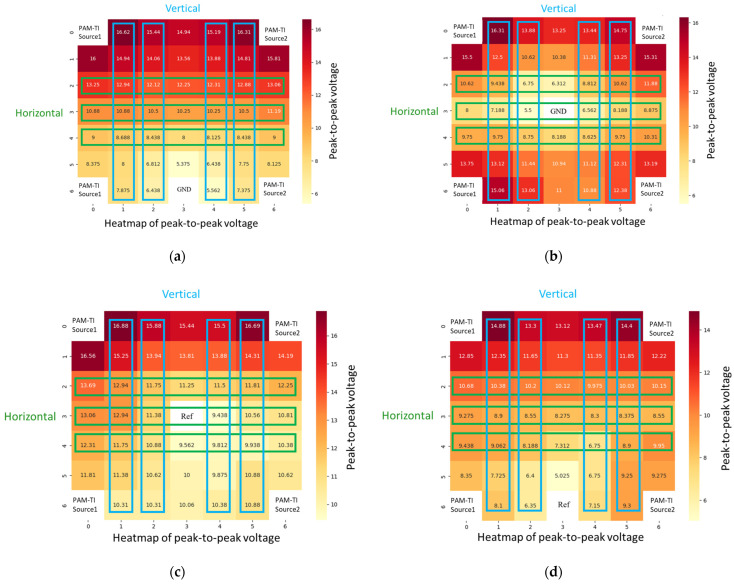
(**a**) Heatmap of peak-to-peak voltage measuring with GND as the center; (**b**) Heatmap of peak-to-peak voltage measuring with GND as the bottom; (**c**) Heatmap of peak-to-peak voltage measuring with REF as the center; (**d**) Heatmap of peak-to-peak voltage measuring with REF as the bottom.

**Figure 12 bioengineering-12-00164-f012:**
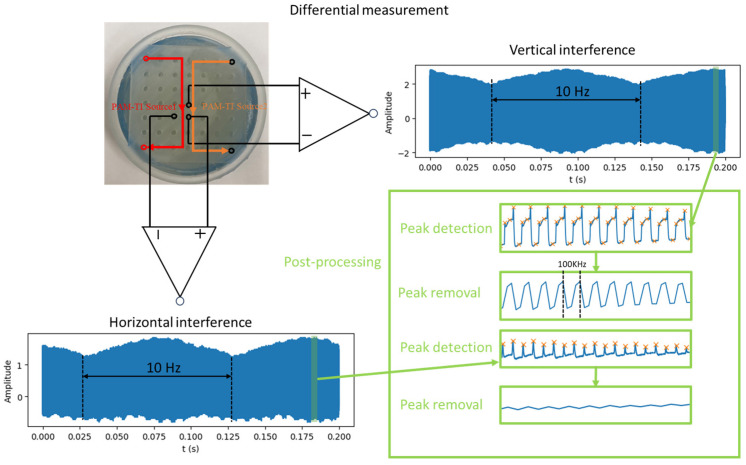
Differential measurement in the vertical and the horizontal directions of the center point.

**Table 1 bioengineering-12-00164-t001:** PWM frequencies and resolutions of the STM32 series.

MCU	Settings	PMW Frequency	Resolution
STM32L4	HCLK (System clock) 80 MHz 80 MHz/(PSC+1)(ARR+1), PSC = 0	80M/796 = 100.502 kHz	126 Hz 125 Hz
		80M/797 = 100.376 kHz	
		80M/798 = 100.25 kHz	
		80M/799 = 100.125 kHz	
		80M/800 = 100 kHz	
STM32F4	HCLK (System clock) 180 MHz 180 MHz/(PSC+1)(ARR+1), PSC = 0	180M/1796 = 100.222 kHz	56 Hz 55 Hz
		180M/1797 = 100.166 kHz	
		180M/1798 = 100.111 kHz	
		180M/1799 = 100.055 kHz	
		180M/1800 = 100 kHz	
STM32F7	HCLK (System clock)216 MHz 216 MHz/(PSC+1)(ARR+1), PSC = 0	216M/2156 = 100.185 kHz	47 Hz 46 Hz
		216M/2157 = 100.139 kHz	
		216M/2158 = 100.092 kHz	
		216M/2159 = 100.046 kHz	
		216M/2160 = 100 kHz	

**Table 2 bioengineering-12-00164-t002:** The summary of neural electrical stimulation device.

Related Works	Charging Current Balance	No Transformer	Stimulation Frequency
De Lima et al. (2002) [21]	No	No	10 Hz
Valente et al. (2012) [22]	Yes	Yes	1 kHz
Lin et al. (2013) [23]	Yes	Yes	400 Hz (2.5-ms period)
Kouzani et al. (2016) [24]	No	Yes	150 Hz
Zhao Zhang et al. (2022) [11]	No	No	2 kHz
Corva et al. (2024) [25]	No	Yes	185 Hz
Proposed stimulator	Yes	Yes	100 kHz

**Table 3 bioengineering-12-00164-t003:** The conductivities and compositions of the electrical simulants for a scalp tissue.

Tissue	Target Conductivity (S/m)	Agar Concentration (g/L)	NaCl Concentration (g/L)
Scalp	0.465	30	2

## Data Availability

The data presented in this study are available on request from the corresponding author.

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
