# Peer review of "A High-Frequency Temporal-Interference Alternative Current Stimulation Device Using Pulse Amplitude Modulation with Push–Pull Current Sources"

_bioengineering, 2025, doi:10.3390/bioengineering12020164_

Round 1
Reviewer 1 Report
Comments and Suggestions for Authors
The introduction mentions existing techniques, but the comparison with PAM-TI is superficial. A more in-depth comparison highlighting the advantages and disadvantages of PAM-TI over existing methods in terms of efficacy, safety, cost, and ease of use is needed. Specifically, how does PAM-TI address the limitations of prior TI methods (like high impedance at lower frequencies)?
The choice of a 100 kHz carrier frequency is presented without strong justification beyond reduced impedance. The manuscript needs to demonstrate a clear advantage of using 100 kHz versus lower frequencies, addressing potential side effects or limitations associated with such a high frequency. Are there any safety concerns related to the high frequency? What are the trade-offs? Is the 100 kHz frequency necessary for deep brain stimulation, or could lower frequencies be sufficient? The authors need to support their claim with evidence or citations that show this higher frequency is significantly beneficial and safe.
The impact of component tolerances on the accuracy of the generated currents needs to be assessed.
Some figures and data presented (e.g., Figure 11) are hard to interpret. The authors need to improve the clarity of the presentation and provide more detailed explanations. Statistical analysis of the experimental results should be included to support the conclusions drawn.
In Table 1, all units kHz are written with capital "k", like KHz, which is wrong and must be kHz.
Author Response
Response to Reviewer 1 Comments
|
||||||||||||||||||||||||
1. Summary |
|
|
||||||||||||||||||||||
Thank you for your valuable feedback. Based on your comments, we have revised and expanded the paper to enhance the completeness and persuasiveness of our discussion. Regarding the comparison between PAM-TI and existing technologies, we have further elaborated in the introduction on its advantages in reducing biological impedance, power consumption, and hardware costs. Additionally, we have included a discussion on how PAM-TI overcomes the frequency and cost limitations of traditional TI technology. In terms of circuit accuracy, we have analyzed the impact of component tolerances on the output accuracy of the current source, providing detailed mathematical derivation. Furthermore, we have discussed the research limitations and future directions, noting that the proposed PAM-TI technique requires further validation through biological experiments to assess the safety and efficacy of neural activation and ensure its reliability in practical applications. These revisions enhance the rigor of the paper. We sincerely appreciate your review and constructive comments. |
||||||||||||||||||||||||
2. Questions for General Evaluation |
Reviewer’s Evaluation |
Response and Revisions |
||||||||||||||||||||||
Does the introduction provide sufficient background and include all relevant references? |
Can be improved |
|
||||||||||||||||||||||
Is the research design appropriate? |
Can be improved |
|
||||||||||||||||||||||
Are the methods adequately described? |
Must be improved |
|
||||||||||||||||||||||
Are the results clearly presented? |
Must be improved |
|
||||||||||||||||||||||
Are the conclusions supported by the results? |
Can be improved |
|
||||||||||||||||||||||
|
|
|
||||||||||||||||||||||
3. Point-by-point response to Comments and Suggestions for Authors |
||||||||||||||||||||||||
Comments 1: The introduction mentions existing techniques, but the comparison with PAM-TI is superficial. A more in-depth comparison highlighting the advantages and disadvantages of PAM-TI over existing methods in terms of efficacy, safety, cost, and ease of use is needed. Specifically, how does PAM-TI address the limitations of prior TI methods (like high impedance at lower frequencies)? |
||||||||||||||||||||||||
Response 1: We appreciate reviewer’s helpful comments. According to previous studies [8,9], higher stimulation frequencies can reduce biological impedance, meaning that at the same stimulation current intensity, stimulation signals with higher frequencies can achieve the desired effect with a lower output voltage. The reduction in output voltage not only enhances the safety of the system but also reduces the supply voltage requirements, thereby decreasing the power consumption and heat dissipation of the output operational amplifier (OPA), making it more suitable for wearable system applications.
Figure 1. Bioimpedance versus frequency [9]. Traditional TI technology uses only 2 kHz and 2.01 kHz sine waves for interference, which still faces the issue of high biological impedance. To further increase the stimulation frequency, this would be constrained by the frequency limit of low-resource embedded systems, resulting in high costs. However, while low-resource embedded systems may struggle to generate high-frequency sine waves, they can easily produce high-frequency Pulse Width Modulation (PWM) signals. Therefore, this paper proposes using PWM signals and peripheral circuits as carriers to perform Pulse Amplitude Modulation (PAM) on the traditional TI signals, thereby effectively increasing the stimulation current frequency and overcoming the technical limitations of traditional TI at high-frequency stimulation. We have revised Section 1 to clearly outline the specific differences between PAM-TI and traditional TI, emphasizing the advantages of PAM-TI in system performance and hardware design. Additionally, we added Section 2.3.2 to analyze the impact of reduced output voltage on the OPA’s power consumption and heat dissipation. The revised content is as follows:
(page 2, line 10) According to previous studies [8,9], higher stimulation frequencies can reduce biological impedance, meaning that at the same stimulation current intensity, stimulation signals with higher frequencies can achieve the desired effect with a lower output voltage. The reduction in output voltage not only enhances the safety of the system but also reduces the supply voltage requirements, thereby decreasing the power consumption and heat dissipation of the output operational amplifier (OPA), making it more suitable for wearable system applications
(page 2, line 39) Ultimately, we propose a high-frequency PAM-TI stimulation system. Compared to the traditional TI stimulation frequency of 2 kHz, PAM-TI can achieve stimulation frequencies of up to 100 kHz, significantly enhancing the system's frequency range and application potential. In terms of efficiency and safety, PAM-TI effectively reduces the operating voltage of the system due to the lower output impedance, which not only improves the system's safety but also reduces power consumption, thereby enhancing overall energy efficiency. This makes it more suitable for prolonged use and wearable applications. In terms of cost, increasing the frequency of traditional TI systems is constrained by the operating frequency of embedded systems, leading to a significant rise in system costs. In contrast, PAM-TI utilizes PWM signals and peripheral circuits for Pulse Amplitude Modulation (PAM), enabling higher-frequency stimulation signals without significantly increasing hardware costs, overcoming the technical and cost limitations of traditional TI systems at high-frequency stimulation.
(page 8, line 8) The power consumption of the OPA, , can be divided into internal power and output power , as expressed by the following equation:
Where is the static current of the OPA, is the supply voltage of the OPA, is the output current of the OPA, and is the output voltage of the OPA. The temperature rise of the OPA component can be estimated using equation (17).
Where is the junction temperature of the OPA, is the ambient temperature, and is the thermal resistance (°C/W). Based on equations (16) and (17), it is evident that PAM-TI has a significant advantage over traditional TI. Due to the higher stimulation frequency of PAM-TI, it effectively reduces the bioimpedance, thereby lowering the operational amplifier (OPA) supply voltage and output voltage . This not only significantly reduces the current source OPA power dissipation , improving the overall system efficiency, but also results in a lower operating temperature under the same cooling conditions, mitigating the risk of overheating and enhancing the stability during prolonged operation.
|
||||||||||||||||||||||||
Comments 2: The choice of a 100 kHz carrier frequency is presented without strong justification beyond reduced impedance. The manuscript needs to demonstrate a clear advantage of using 100 kHz versus lower frequencies, addressing potential side effects or limitations associated with such a high frequency. Are there any safety concerns related to the high frequency? What are the trade-offs? Is the 100 kHz frequency necessary for deep brain stimulation, or could lower frequencies be sufficient? The authors need to support their claim with evidence or citations that show this higher frequency is significantly beneficial and safe. |
||||||||||||||||||||||||
Response 2: Thank you for your valuable feedback. This paper proposes PAM-TI, which utilizes high-frequency stimulation signals to reduce biological impedance. Additionally, we have developed an EIP circuit that integrates the stimulation circuit into the electrode, allowing the stimulation current to flow directly into biological tissues, thereby minimizing the impact of wiring on high-frequency stimulation currents. However, the safety and neural stimulation efficiency of PAM-TI require further investigation. We have included discussions on limitations and future research directions in Section 4. The relevant content is as follows:
(page 13, line 23) In this paper, we introduce PAM-TI and analyze its advantages over traditional TI, partic-ularly in increasing stimulation frequency, reducing biological impedance, enhancing system efficiency, and improving heat dissipation. However, the impact of PAM-TI on safety and neural stimulation efficiency requires further validation and evaluation. In fu-ture research, we will conduct biological experiments to further assess the effectiveness of PAM-TI in real biological environments and ensure its safety and efficacy for clinical ap-plications.
|
||||||||||||||||||||||||
Comments 3: The impact of component tolerances on the accuracy of the generated currents needs to be assessed. |
||||||||||||||||||||||||
Response 3: Thake you for your insightful comment. We have revised Section 2.3.3 and analyzed the impact of component tolerances on the accuracy of the current source output current. The revised content is as follows:
(page 7, line 7)
Considering the impact of component errors on the output current accuracy of the Howland current source circuit, the following derivation is performed: Based on the virtual ground characteristic of the OPA, equation (7) can be obtained.
Where , , , and represent the resistance errors of resistors , , , and , respectively, and . Simplifying equation (7) yields equation (8).
To simplify the analysis, set and , which gives:
Equation (9) can be simplified through basic algebraic operations to obtain equation (10).
Since , , and will be much larger than the error terms, equation (10) can be approximated as:
Simplifying (11) gives:
Based on (12), the output current is:
Through equation (13), the output current can be separated into the ideal output current and the output current error .
Based on relation (17), the output current error can be designed according to the application requirements.
Where is the maximum acceptable error percentage in the application.
|
||||||||||||||||||||||||
Comments 4: Some figures and data presented (e.g., Figure 11) are hard to interpret. The authors need to improve the clarity of the presentation and provide more detailed explanations. Statistical analysis of the experimental results should be included to support the conclusions drawn. |
||||||||||||||||||||||||
Response 4: We have revised and clarified the description of Fig. 11 in Section 3.
(page 11, line 2) To verify the current directionality of the push-pull current source, PAM-TI Source1 was connected to the upper left/lower left of the sampling box, and PAM-TI Source2 was connected to the upper right/lower right of the sampling box. A probe was then used to record the voltage generated by the current flowing through the tissue-mimicking phan-tom, allowing us to observe the current flow distribution via the voltage difference. Peak-to-peak voltage (Vpp) measurements were used to generate a heatmap. As shown in Figure 11, a significant voltage difference was observed in the vertical direction, while the voltage difference in the horizontal direction was less pronounced. This experiment aimed to confirm the influence of REF (where only the ground of the measuring probe is con-nected to the sampling box) or GND (where both the ground of the measuring probe and the stimulator system are connected to the sampling box) on the direction of the current. Therefore, REF or GND was placed at the middle or bottom-center position. It was ob-served that a slight voltage depression occurred around the reference point or ground, but the vertical and horizontal current directionality remained intact. These observations val-idate the feasibility of the push-pull current source proposed in this study.
|
||||||||||||||||||||||||
Comments 5: In Table 1, all units kHz are written with capital "k", like KHz, which is wrong and must be kHz. |
||||||||||||||||||||||||
Response 5: We have corrected the units in Table 1.
|
Reviewer 2 Report
Comments and Suggestions for Authors
This manuscript mainly focuses on the realization of the device, which implements the Pulse Amplitude Modulation Temporal Interference (PAM-TI) method, to enhance the stimulation depth and efficiency for DBS application. The main novelty comes from the combination of a 100 kHz carrier frequency, which can reduce the electrode-tissue impedance and conserve more energy for neural stimulation, according to the authors’ claim.
Currently, there are several major flaws that should be addressed, as listed below, before the manuscript can be published.
1. In page 3 and Figure 1(c), the dry pin-type spring-loaded electrodes were used to overcome interference caused by hair. This electrode is quite different from the normal EMG or TMS wet electrodes, which normally has a hydrogel to achieve good contact with the skin. The authors should provide references that can support the application and performance of this dry pin-type spring-loaded electrodes for transcranial stimulations.
2. Figure 2 is not well arranged. The sub-figures should be aligned with label followed.
3. The main motivation of this manuscript is that a higher carrier frequency can reduce the electrode-tissue impedance and thus conserve more energy for stimulation. Since the circuit configuration of the skin-tissue interface is an RC circuit, surely a higher frequency can reduce this interface impedance, but a higher frequency will also induce a higher decay coefficient of the signal in tissue. Thus it is not so convincing that this higher carrier frequency can really enhance the efficiency. Meanwhile, in the part of results, the authors also do not provide a comparison of the new PAM-TI method and the conventional TI method. Therefore, the major claim of this manuscript is not well supported.
4. In this manuscript, the authors claim that the PAM circuit was integrated with the electrodes, which avoid a long transmission line that will cause signal attenuation. And the system is transformer-free, which makes it more suitable for wearable applications. Then for such a highly integrated system, what is its power consumption and whether it has heat dissipation issue?
Comments on the Quality of English Language
Can be improved accordingly.
Author Response
Response to Reviewer 2 Comments
|
||||||||||
1. Summary |
|
|
||||||||
Thank you for your valuable feedback. Based on your comments, we have revised and supplemented the paper, incorporating relevant research to support our claims regarding the application of dry spring-loaded electrodes. While dry electrodes exhibit higher contact impedance, they provide greater resistance to environmental noise and eliminate the need for conductive gel, making them advantageous for practical applications. Additionally, we have reorganized the figures and tables to enhance clarity and readability. To further compare PAM-TI with traditional TI, we have highlighted its advantages in reducing biological impedance, improving efficiency, and lowering power consumption. Moreover, we have included an analysis of how reduced output voltage impacts OPA power consumption and heat dissipation. Furthermore, we have discussed the research limitations and future directions, noting that the proposed PAM-TI technique requires further validation through biological experiments to assess the safety and efficacy of neural activation and ensure its reliability in practical applications. These revisions enhance the rigor and comprehensiveness of the manuscript. We sincerely appreciate your meticulous review and valuable guidance. |
||||||||||
2. Questions for General Evaluation |
Reviewer’s Evaluation |
Response and Revisions |
||||||||
Does the introduction provide sufficient background and include all relevant references? |
Yes |
|
||||||||
Is the research design appropriate? |
Can be improved |
|
||||||||
Are the methods adequately described? |
Yes |
|
||||||||
Are the results clearly presented? |
|
|
||||||||
Are the conclusions supported by the results? |
Can be improved |
|
||||||||
|
|
|
||||||||
3. Point-by-point response to Comments and Suggestions for Authors |
||||||||||
Comments 1: In page 3 and Figure 1(c), the dry pin-type spring-loaded electrodes were used to overcome interference caused by hair. This electrode is quite different from the normal EMG or TMS wet electrodes, which normally has a hydrogel to achieve good contact with the skin. The authors should provide references that can support the application and performance of this dry pin-type spring-loaded electrodes for transcranial stimulations. |
||||||||||
Response 1: Thake you for your insightful comment. Hinrichs et al. [18] demonstrated that dry electrodes produce measurement results comparable to wet electrodes. This indicates that dry and wet electrodes can establish valid connections on the skin. Despite their higher contact impedance, dry electrodes exhibit greater robustness against environmental noise. Since dry electrodes are more convenient to use and do not require conductive gels, they are generally preferred by users, making them highly promising for various applications. In this paper, the PAM-TI stimulation frequency is significantly higher than traditional TI, effectively mitigating the impact of the higher contact impedance associated with dry electrodes. Furthermore, the proposed EIP circuit is compatible with dry and wet electrodes, as illustrated in the figure below.
We have revised the description of the proposed EIP circuit and incorporated information about dry and wet electrodes to more comprehensively explain the advantages of dry electrodes. The added content is as follows:
(page 4, line 15) The EIP circuit designed in this paper offers high compatibility and can be used with either dry or wet electrodes, which can be selected according to the specific application needs, as shown in Figure 2(c) and Figure 2(d). Previous studies [18,19] have indicated that both dry and wet electrodes can establish effective electrical contact on the skin. Although dry electrodes result in higher contact impedance, they are generally preferred by users since they do not require conductive gels, highlighting their potential for practical applications. In the context of this paper, the PAM-TI stimulation frequency is much higher than that of traditional TI, which further mitigates the impact of higher contact impedance in dry electrodes, ensuring the stability of the stimulation signal and the applicability of the system.
Figure 2. (a) Integrated power supply main control board. (b) Electrode-integrated PAM circuit. (c) EIP circuit with dry pin-type spring-loaded electrode. (d) EIP circuit with wet electrode. |
||||||||||
Comments 2: Figure 2 is not well arranged. The sub-figures should be aligned with label followed. |
||||||||||
Response 2: We have revised Figure 3 (formerly Figure 2) to make it clearer, with the subplots arranged more neatly.
|
||||||||||
Comments 3: The main motivation of this manuscript is that a higher carrier frequency can reduce the electrode-tissue impedance and thus conserve more energy for stimulation. Since the circuit configuration of the skin-tissue interface is an RC circuit, surely a higher frequency can reduce this interface impedance, but a higher frequency will also induce a higher decay coefficient of the signal in tissue. Thus it is not so convincing that this higher carrier frequency can really enhance the efficiency. Meanwhile, in the part of results, the authors also do not provide a comparison of the new PAM-TI method and the conventional TI method. Therefore, the major claim of this manuscript is not well supported. |
||||||||||
Response 3: Thake you for your insightful comment. We have revised Section 1 to clearly outline the specific differences between PAM-TI and traditional TI, emphasizing the advantages of PAM-TI in system performance and hardware design. Additionally, we added Section 2.3.2 to analyze the impact of reduced output voltage on the OPA’s power consumption and heat dissipation. The revised content is as follows:
(page 2, line 39) Ultimately, we propose a high-frequency PAM-TI stimulation system. Compared to the traditional TI stimulation frequency of 2 kHz, PAM-TI can achieve stimulation frequencies of up to 100 kHz, significantly enhancing the system's frequency range and application potential. In terms of efficiency and safety, PAM-TI effectively reduces the operating voltage of the system due to the lower output impedance, which not only improves the system's safety but also reduces power consumption, thereby enhancing overall energy efficiency. This makes it more suitable for prolonged use and wearable applications. In terms of cost, increasing the frequency of traditional TI systems is constrained by the operating frequency of embedded systems, leading to a significant rise in system costs. In contrast, PAM-TI utilizes PWM signals and peripheral circuits for Pulse Amplitude Modulation (PAM), enabling higher-frequency stimulation signals without significantly increasing hardware costs, overcoming the technical and cost limitations of traditional TI systems at high-frequency stimulation.
(page 8, line 8) The power consumption of the OPA, , can be divided into internal power and output power , as expressed by the following equation:
Where is the static current of the OPA, is the supply voltage of the OPA, is the output current of the OPA, and is the output voltage of the OPA. The temperature rise of the OPA component can be estimated using equation (17).
Where is the junction temperature of the OPA, is the ambient temperature, and is the thermal resistance (°C/W). Based on equations (16) and (17), it is evident that PAM-TI has a significant advantage over traditional TI. Due to the higher stimulation frequency of PAM-TI, it effectively reduces the bioimpedance, thereby lowering the operational amplifier (OPA) supply voltage and output voltage . This not only significantly reduces the current source OPA power dissipation , improving the overall system efficiency, but also results in a lower operating temperature under the same cooling conditions, mitigating the risk of overheating and enhancing the stability during prolonged operation.
This paper proposes PAM-TI, which utilizes high-frequency stimulation signals to reduce biological impedance. Additionally, we have developed an EIP circuit that integrates the stimulation circuit into the electrode, allowing the stimulation current to flow directly into biological tissues, thereby minimizing the impact of wiring on high-frequency stimulation currents. However, the safety and neural stimulation efficiency of PAM-TI require further investigation. We have included discussions on limitations and future research in Section 4. The relevant content is as follows:
(page 13, line 23) In this paper, we introduce PAM-TI and analyze its advantages over traditional TI, particularly in increasing stimulation frequency, reducing biological impedance, enhancing system efficiency, and improving heat dissipation. However, the impact of PAM-TI on safety and neural stimulation efficiency requires further validation and evaluation. In future research, we will conduct biological experiments to further assess the effectiveness of PAM-TI in real biological environments and ensure its safety and efficacy for clinical ap-plications.
|
||||||||||
Comments 4: In this manuscript, the authors claim that the PAM circuit was integrated with the electrodes, which avoid a long transmission line that will cause signal attenuation. And the system is transformer-free, which makes it more suitable for wearable applications. Then for such a highly integrated system, what is its power consumption and whether it has heat dissipation issue? |
||||||||||
Response 4: The total power consumption of the entire system (including the power converter, MCU, stimulation circuit, human-machine interface, etc.) is currently only 5W, demonstrating the feasibility of the system for wearable device applications. Additionally, regarding the heat dissipation issue, in Section 2, Materials and Methods, we have added an analysis of the impact of the reduced output voltage on OPA power dissipation and heat dissipation, highlighting the advantages of PAM-TI in system performance and hardware design. The following are the newly added contents: (page 8, line 8) The power consumption of the OPA, , can be divided into internal power and output power , as expressed by the following equation:
Where is the static current of the OPA, is the supply voltage of the OPA, is the output current of the OPA, and is the output voltage of the OPA. The temperature rise of the OPA component can be estimated using equation (17).
Where is the junction temperature of the OPA, is the ambient temperature, and is the thermal resistance (°C/W). Based on equations (16) and (17), it is evident that PAM-TI has a significant advantage over traditional TI. Due to the higher stimulation frequency of PAM-TI, it effectively reduces the bioimpedance, thereby lowering the operational amplifier (OPA) supply voltage and output voltage . This not only significantly reduces the current source OPA power dissipation , improving the overall system efficiency, but also results in a lower operating temperature under the same cooling conditions, mitigating the risk of overheating and enhancing the stability during prolonged operation. |

Reviewer 3 Report
Comments and Suggestions for Authors The article presents a profoundly insightful study, proposing an innovative approach to enhancing non-invasive Brain-Computer Interface (BCI) technology for brain wave resonant biomedical instrumentation. In many ways, it echoes the well-established eleventh harmonics or Hemi-Sync approach and the Solfeggio frequencies, which utilize acoustic frequencies to activate and control different brain regions through electromagnetic wave alternation. This can optimize biomaterial activation and synchronize brain waves and hemisphere patterns. The current version of the paper is both broad and in-depth, offering a detailed overview of the techniques while focusing on the author's proposed approach to optimizing the non-invasive interface for brain transduction and induction through frequencies. In light of the author's groundbreaking contribution, I strongly advocate for expedited publication. This remarkable study constitutes a seminal work in the field, warranting prompt dissemination. In my expert opinion, this article is one of the most outstanding contributions to this area in recent years.Author Response
Response to Reviewer 3 Comments
|
||
1. Summary |
|
|
We sincerely appreciate your thoughtful and encouraging feedback. Your recognition of our work as a significant contribution to the field of non-invasive brain stimulation and Brain-Computer Interface (BCI) technology is truly gratifying. We are especially honored by your endorsement of our proposed PAM-TI technique and its potential impact on optimizing brain transduction and induction. Thank you for your time, expertise, and support in reviewing our work.
|
||
2. Questions for General Evaluation |
Reviewer’s Evaluation |
Response and Revisions |
Does the introduction provide sufficient background and include all relevant references? |
Yes |
|
Is the research design appropriate? |
Yes |
|
Are the methods adequately described? |
Yes |
|
Are the results clearly presented? |
Yes |
|
Are the conclusions supported by the results? |
Yes |
|
|
|
|
3. Point-by-point response to Comments and Suggestions for Authors |
||
Comments 1: The article presents a profoundly insightful study, proposing an innovative approach to enhancing non-invasive Brain-Computer Interface (BCI) technology for brain wave resonant biomedical instrumentation. In many ways, it echoes the well-established eleventh harmonics or Hemi-Sync approach and the Solfeggio frequencies, which utilize acoustic frequencies to activate and control different brain regions through electromagnetic wave alternation. This can optimize biomaterial activation and synchronize brain waves and hemisphere patterns. The current version of the paper is both broad and in-depth, offering a detailed overview of the techniques while focusing on the author's proposed approach to optimizing the non-invasive interface for brain transduction and induction through frequencies. In light of the author's groundbreaking contribution, I strongly advocate for expedited publication. This remarkable study constitutes a seminal work in the field, warranting prompt dissemination. In my expert opinion, this article is one of the most outstanding contributions to this area in recent years. |
||
Response 1: We sincerely appreciate your thoughtful and encouraging feedback. Your recognition of our work as a significant contribution to the field of non-invasive brain stimulation and Brain-Computer Interface (BCI) technology is truly gratifying. We are especially honored by your endorsement of our proposed PAM-TI technique and its potential impact on optimizing brain transduction and induction. Thank you for your time, expertise, and support in reviewing our work. |
Round 2
Reviewer 1 Report
Comments and Suggestions for Authors
Dear Authors,
Excellent addressing of all comments! The quality of the presentation is very high now and, thus, easier to read. I have no more comments and suggest acceptance in the present form.
Reviewer 2 Report
Comments and Suggestions for Authors
All questions have been replied.